# Lost in Simulation: LLM-Simulated Users are Unreliable Proxies for Human Users in Agentic Evaluations

**Preethi Seshadri**[1], **Samuel Cahyawijaya**[2], **Ayomide Odumakinde**[2], **Sameer Singh**[1],
**Seraphina Goldfarb-Tarrant**[2]
[1]UC Irvine     [2]Cohere
{preethis, sameer}@uci.edu
{samuelcahyawijaya, ayomideodumakinde, seraphina}@cohere.com

## ABSTRACT

Agentic benchmarks increasingly rely on LLM-simulated users to scalably evaluate agent performance, yet the robustness, validity, and fairness of this approach remain unexamined. Through a user study with participants across the United States, India, Kenya, and Nigeria, we investigate whether LLM-simulated users serve as reliable proxies for real human users in evaluating agents on $\tau$-Bench retail tasks. We find that user simulation lacks robustness, with agent success rates varying up to 9 percentage points across different user LLMs. Furthermore, evaluations using simulated users exhibit systematic miscalibration, underestimating agent performance on challenging tasks and overestimating it on moderately difficult ones. African American Vernacular English (AAVE) speakers experience consistently worse success rates and calibration errors than Standard American English (SAE) speakers, with disparities compounding significantly with age. We also find simulated users to be a differentially effective proxy for different populations, performing worst for AAVE and Indian English speakers. Additionally, simulated users introduce conversational artifacts and surface different failure patterns than human users. These findings demonstrate that current evaluation practices risk misrepresenting agent capabilities and may obscure deployment challenges that emerge when agents interact with diverse users in the wild.

## 1 INTRODUCTION

AI agents designed to assist with everyday tasks such as travel reservations, order management, and appointment scheduling are becoming increasingly prevalent (Xie et al., 2024; Zhou et al., 2024; CNBC, 2025), but present significant challenges to effective evaluation. Agentic benchmarks have needed to evolve beyond static question-answering and other single-turn formats to capture the dynamic, multi-turn nature of real user interactions (Chang et al., 2025; Deshpande et al., 2025). Many recent works have proposed benchmarks that reflect this shift and instead measure sustained, context-aware interaction (Barres et al., 2025; Shao et al., 2025; Wang et al., 2025a; Xu et al., 2025; Yao et al., 2025). These benchmarks improve complexity and ecological validity over prior static evaluations by requiring agents to demonstrate a range of capabilities, including conversing naturally and coherently with users, adhering to policies, following instructions closely, and making appropriate tool calls over multiple turns. To facilitate automated and scalable evaluation, these benchmarks typically simulate conversations between an LLM agent and a user, where the "user" is an LLM (Ivey et al., 2024; He et al., 2025; Mehri et al., 2025).

While this approach reduces the cost and operational overhead of human evaluation, it raises critical questions about the **robustness**, **validity**, and **fairness** of user simulation. First, evaluations typically rely on a single user simulation model, yet results may vary across different user LLMs (**robustness**). Second, without validation with actual users (Salaudeen et al., 2025), it remains unclear whether interactions between agents and LLM-simulated users accurately reflect and predict interactions between agents and real people (**validity**, Figure 1). If simulated users systematically differ from actual users in their interaction patterns (Yoon et al., 2024), benchmarks may provide a misleading

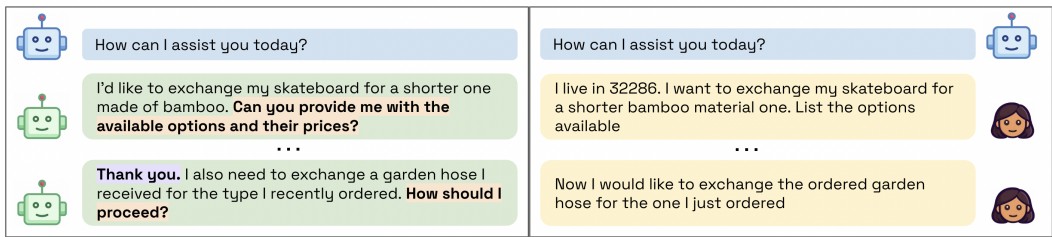

Figure 1: Conversational snippets between an agent and simulated (top) vs. human user (bottom) on the same task. Simulated users exhibit increased question-asking and politeness compared to human users.

picture of agent capabilities due to *miscalibration* (i.e., simulated results do not reliably predict real user outcomes).

Third, these evaluations often treat users as a homogeneous group, overlooking variation in how people communicate and interact with AI systems (Haoyue & Cho, 2024; Liu et al., 2024; Bassignana et al., 2025). In practice, users differ widely in their communication styles, linguistic backgrounds, and cultural norms (Pawar et al., 2025; Qiu et al., 2025). For example, even in a simple retail assistance scenario, users might vary along dimensions such as formality, verbosity, and politeness norms—but it remains unclear how much this diversity meaningfully impacts agent performance and task success (Truong et al., 2025). Without further investigation, simulated users may approximate some populations better than others, resulting in non-uniform calibration errors that advantage or disadvantage certain user groups (**fairness**).

Despite the widespread adoption of user simulation, prior work has overlooked validation against real human interactions in agentic benchmarks. In this paper, we address this gap by conducting a user study with participants from the United States, India, Kenya, and Nigeria to directly evaluate user simulation as a proxy for actual users. Specifically, we ask the following questions:

1. **Robustness:** How consistent are agentic evaluations across different user simulation LLMs?
2. **Validity:** Do simulated users serve as reliable proxies for human users in agentic evaluations?
3. **Fairness:** How does human-agent performance vary across different user groups, and does user simulation represent certain groups better than others?

Using $\tau$-Bench retail tasks (Yao et al., 2025) as a case study, we find that user simulation lacks robustness, and systematically misestimates performance with human users by underestimating success on the most challenging tasks while overestimating outcomes on moderately difficult scenarios. More critically, simulated users exhibit notable demographic biases and perform particularly poorly as proxies for African American Vernacular English speakers, with disparities compounding with age. Simulated users also introduce artificial conversational artifacts such as heightened question-asking and politeness (Figure 1). Together, these findings call into question the validity of user simulation as a stand-alone evaluation paradigm and underscore the need for more robust and fair approaches to agentic evaluation.

## 2 RELATED WORK

**User Simulation in Interactive Settings**   Dou et al. (2025) and Wang et al. (2025b) investigate user simulation in tasks such as math tutoring and daily planning, but primarily focus on conversational characteristics (e.g., politeness) and behavioral realism (e.g., Turing-style tests). Additionally, Dou et al. (2025) optimize alignment with human ratings using simulated user profiles. Lu et al. (2025) analyze simulation fidelity by measuring how accurately simulated users replicate human intermediate steps in real-world online shopping sessions, and find substantial deviations between simulated and actual user action sequences. None of these works examine the robustness of agentic evaluations across different user simulation models, nor do they consider fairness implications across different user groups. Finally, while Zhu et al. (2025) study the outcome and task validity of agentic benchmark results, they do not consider the validity of user simulation.

**Demographic Skews in NLP Datasets and Models**   Several previous works highlight that datasets and models exhibit systematic skews toward specific demographic perspectives. Research on annotator disagreement reveals that perceptions of safety, offensiveness, and toxicity meaningfully vary along demographic axes like race, gender, and political affiliation (Sap et al., 2022; Lee et al., 2023; Prabhakaran et al., 2024). These patterns of disagreement reflect broader issues of positionality—both Santy et al. (2023) and Lee et al. (2024) show that NLP datasets and models tend to align predominantly with Western, educated, and Anglosphere populations. Similarly, LLMs have been found to reflect the opinions of Western countries (Durmus et al., 2024; Cahyawijaya et al., 2025) as well as wealthy and liberal groups (Santurkar et al., 2023), and align more closely with White annotators than Black or Asian groups on subjective tasks like politeness (Sun et al., 2025). Attempts to broaden model inclusivity through sociodemographic prompting have shown mixed results, often failing to consistently improve alignment or relying on harmful stereotypes (Durmus et al., 2024; Sun et al., 2025). Overall, our work builds on this line of research by examining demographic differences in agentic settings.

## 3 METHODOLOGY

### 3.1 BENCHMARK BACKGROUND

We use $\tau$-Bench (Yao et al., 2025) as the testbed for our evaluations, since it is a well-known and widely-adopted benchmark for agentic tool use.[1] The benchmark is designed to capture how well AI agents perform in real-world, interactive customer service scenarios. Each task involves collaboration between an agent and a simulated user: the user receives task instructions with specific objectives that guide their conversation with the agent, while the agent must use tool calling to interact with realistic databases, adhere to domain-specific policies, and gather necessary information from the user.

The task is considered successful if (1) the final database state is identical to the unique ground truth outcome (i.e., the sequence of required actions) and (2) the agent's responses convey all necessary information requested in the task instructions, which is evaluated automatically using substring matching against ground truth annotations. In total, $\tau$-Bench contains 115 retail tasks (e.g., modifying pending orders or returning delivered orders) and 50 airline tasks (e.g., booking, modifying, or canceling reservations).

### 3.2 BENCHMARK ADAPTATION

We focus on $\tau$-Bench retail tasks to enable more systematic coverage within a single domain. We apply preprocessing steps to ensure that neither simulated nor human users are influenced by identity and behavioral cues in the instructions when completing tasks (see Appendix A.5).

Given the large number of retail tasks, we sample a subset based on difficulty to ensure balanced coverage across task complexities. Since $\tau$-Bench does not provide difficulty labels, we compute a model-based notion of difficulty by running the benchmark 5 times using GPT-4o as both the agent and user LLMs and measuring the task success rate over 5 runs (i.e., the percentage of times a given task is completed successfully). With five evaluation runs per task, success rates naturally fall into six discrete levels (0/5, 1/5, 2/5, 3/5, 4/5, 5/5 = 0%, 20%, 40%, 60%, 80%, 100%). We use GPT-4o because it is used in the $\tau$-Bench paper and does not exhibit contamination issues that newer models face.[2] We then select 3 tasks for each difficulty level (18 total) to balance response coverage per task with breadth across difficulty levels.

### 3.3 USER STUDY

To assess whether LLM-simulated users serve as effective proxies for real and diverse users, we conduct a user study with participants from the United States, India, Kenya, and Nigeria. Since $\tau$-Bench tasks are in English, we select countries with large English-speaking populations that also

---

[1]$\tau$-Bench Leaderboard: `https://taubench.com/#leaderboard`

[2]GPT-4o (May 2024) predates $\tau$-Bench (June 2024), avoiding contamination issues present in newer models. We considered Sonnet 3.5 (also used in the original paper) but it was retired in October 2025.

provide geographical and linguistic diversity.[3] We recruit participants primarily through Prolific, except in Nigeria, where we use snowball sampling due to limited platform availability. All participants self-identify as at least proficient in English.

Each participant completes 4 randomly assigned tasks from our pool of 18, presented in randomized order: 2 from higher difficulty levels (0-40% success rate) and 2 from lower difficulty levels (60-100% success rate). The agent model remains GPT-4o throughout all interactions. Participants are shown task instructions (see Appendix A.6) and asked to complete all mentioned requests by interacting with the agent through a Streamlit chat interface. After finishing each conversation, they click an "End Conversation" button to proceed to the next task. In total, the expected time to complete all 4 tasks is 35-40 minutes.

Participants provide demographic information including education level, AI familiarity, and frequency of AI tool usage. For US participants, we screen for White Standard American English (SAE) speakers and Black African American Vernacular English (AAVE) speakers based on self-reported race and dialect, as these groups are commonly studied in AI fairness research (Sap et al., 2019; Groenwold et al., 2020). We also stratify US participants by age (18-34, 35-54, and 55+) to capture potential differences in technology experience (Pew Research Center, 2025). Due to participant availability constraints, we only recruit from the 18-34 age group for other countries. In total, we recruit ~40 participants per age × dialect/country group.[4]

## 3.4 EVALUATION METRICS

**Success Rate**    To recap, a task completion in $\tau$-Bench is considered successful ($reward = 1$) if and only if the agent correctly executes all required actions and its responses convey all information specified in the instructions. We define success rate as the percentage of tasks that are successfully completed. We apply the same automated evaluation procedure used in $\tau$-Bench to both human and simulated user interactions. Success rate is averaged across difficulty levels to obtain a single value.

**Expected Calibration Error (ECE)**    We adapt *Expected Calibration Error (ECE)*, commonly used for assessing confidence calibration in probabilistic classifiers, to quantify how well simulated users serve as proxies for human users. While traditional ECE evaluates whether a model's predicted probabilities match true observed outcomes, we use an ECE-style formulation to measure whether agent success rates with simulated users align with agent success rates with real users across task difficulty levels. Just as traditional ECE measures whether predicted confidences are calibrated to observed outcomes (Guo et al., 2017), our metric measures calibration between agent performance distributions under simulated and real user conditions.

Let $s_i^{(\text{LLM})}$ denote success rate with LLM-simulated users and $s_i^{(\text{Human})}$ denote success rate with human users at difficulty level $i$. Let $w_i$ denote the proportion of human task completions at level $i$, with $\sum_i w_i = 1$. We define:

$$\text{ECE}_{\text{Human–LLM}} = \sum_{i=1}^{M} w_i \left| s_i^{(\text{Human})} - s_i^{(\text{LLM})} \right|$$

This metric captures the weighted average absolute deviation between agent performance when interacting with simulated users vs. real users across $M$ difficulty levels, with lower values indicating better calibration. If simulated users are perfectly calibrated to real users, $ECE_{\text{Human–LLM}} = 0$.

As a reminder, we partition tasks into six difficulty levels based on agent success rates with simulated users across five runs. We set $w_i$ proportional to the number of human task completions at level $i$. We multiply $ECE_{\text{Human–LLM}}$ by 100 to report values as percentages.

---

[3]https://en.wikipedia.org/wiki/List_of_countries_by_English-speaking_population

[4]With the exception of the AAVE 55+ group, for which we were only able to recruit 22 participants.

## 4 RESULTS

### 4.1 ROBUSTNESS

We first evaluate the robustness of user simulation by examining how success rates vary with different user simulation models, which is largely overlooked in current evaluations.[5] We find that changing just the user LLM while keeping the agent LLM fixed (GPT-4o) can provide different depictions of agent performance.

As shown in Table 1, GPT-4o, Sonnet 3.7, and Kimi-K2-Thinking show overlapping intervals, clustering around $67$-$71\%$ success rates. However, there is nearly a 9 percentage point difference in success rates between Sonnet 3.7 and Sonnet 4.5 as the user model.[6] While Sonnet 3.7 is generally considered a stronger model than GPT-4o,[7] using GPT-4o as the user model yields a slightly higher success rate ($67.8$ vs. $67.0$) and lower standard deviation ($1.2$ vs. $3.3$), suggesting that closer

| User Model | Success Rate (%) |
|---|---|
| GPT-4o | $67.8 \pm 1.2$ |
| Sonnet 3.7 | $67.0 \pm 3.3$ |
| Sonnet 4.5 | $75.9 \pm 3.5$ |
| Kimi-K2-Thinking | $71.3 \pm 1.9$ |

Table 1: $\tau$-Bench success rate (%) on retail tasks ($n = 115$).

alignment between agent and user LLMs may lead to more stable evaluation outcomes. Overall, the sensitivity of agent performance to user model choice raises concerns about the reliability of single-model user simulations and underscores the need for reporting results across multiple user models to establish robustness.

### 4.2 VALIDITY

We now examine the validity of user simulation and first consider simulated users as a proxy for human users in the United States. Since prior work has shown that LLMs exhibit a Western, Anglo-centric bias (Tao et al., 2024; Wang et al., 2024; Agarwal et al., 2025), LLM-simulated users might be more representative or better calibrated to users in the US than to users in non-Western countries. Therefore, we assess whether simulated users are well-calibrated to human users in the setting where we expect the strongest alignment.

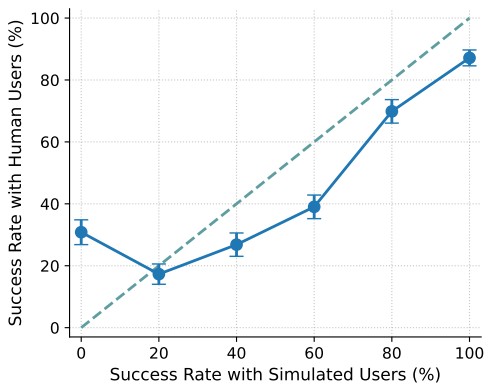

Figure 2: Success rate with human users vs. LLM-simulated users for United States participants, with an $ECE_{\text{Human–LLM}}$ of $15.1$. Error bars indicate $\pm 1$ SD.

We find that agents achieve a $45.2\%$ success rate with US participants and an $ECE_{\text{Human–LLM}}$ of $15.1$, indicating substantial miscalibration even in this setting. Calibration errors are not uniform across task difficulty. As shown in Figure 2, the calibration gap is most pronounced for the 1st ($0\%$) and 4th ($60\%$) difficulty bins with $ECE_{\text{Human–LLM}} = 25.9$ across the two bins. These results indicate that evaluations with simulated users underestimate agent success on the hardest tasks (success with human users: $30.8\%$) while overestimating it on moderate tasks (success with human users: $39.0\%$).

### 4.3 FAIRNESS

To assess whether these findings are consistent across different user groups, we now partition our US results by English dialect (Standard American English and African American Vernacular English)

---

[5]For robustness, we vary the user LLM and evaluate on all retail tasks; for validity and fairness analyses, we evaluate with human users on a difficulty-balanced subset of tasks.

[6]It is difficult to disentangle increased model capability from potential data contamination.

[7]https://lmarena.ai/leaderboard/text

| Age Group | Success Rate (↑) | ECE (↓) |
|---|---|---|
| **SAE** | | |
| All | 50.6 | 11.7 |
| 18–34 | 49.2 | 13.0 |
| 35–54 | 52.2 | 11.3 |
| 55+ | 52.1 | 14.5 |
| **AAVE** | | |
| All | 39.4 | 20.3 |
| 18–34 | 41.0 | 18.9 |
| 35–54 | 39.9 | 21.6 |
| 55+ | 33.4 | 20.5 |

Table 2: Success Rate (%) and Expected Calibration Error ($ECE_{\text{Human–LLM}}$) for Standard American English (SAE) and African American Vernacular English (AAVE) speaking participants, split by age group.

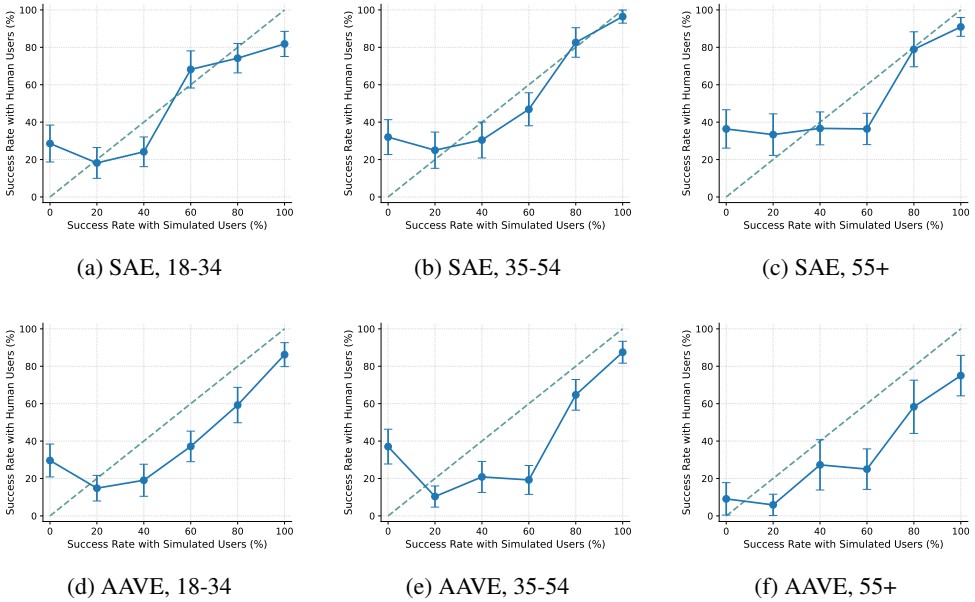

Figure 3: Success rate with human users vs. LLM-simulated users for SAE (top) and AAVE (bottom) participants across different age groups (18-34, 35-54, and 55+). Note: The x-axis (success rate with simulated users) remains the same for all groups.

and age group (18-34, 35-54, and 55+), and expand our analysis to three non-Western countries with high English-speaking populations (India, Kenya, and Nigeria).

### 4.3.1 DIALECT AND AGE (UNITED STATES)

Starting with US participants, we previously saw that agents achieve a $45.2\%$ success rate and an $ECE_{\text{Human–LLM}}$ of $15.1$. When further breaking this down by dialect, we observe notable disparities in both performance and calibration, as shown in Table 2 and Figure 3. A Generalized Estimating Equations (GEE) model accounting for age, education, AI experience, AI usage, and task difficulty confirms a statistically significant dialect disparity ($\beta = 0.61$, $p < 0.001$). Agents exhibit a success rate of $50.6\%$ with an $ECE_{\text{Human–LLM}}$ of $11.7$ for SAE participants vs. a success rate of $39.4\%$ with an $ECE_{\text{Human–LLM}}$ of $20.3$ for AAVE participants. For AAVE participants, agents perform worse (11.2 percentage point decrease in success rate) and simulated users are more poorly calibrated (8.6 percentage point increase in ECE). In practice, such differences in performance and user simulation reliability could lead to disparities in the quality of retail assistance and the ease of interactions.

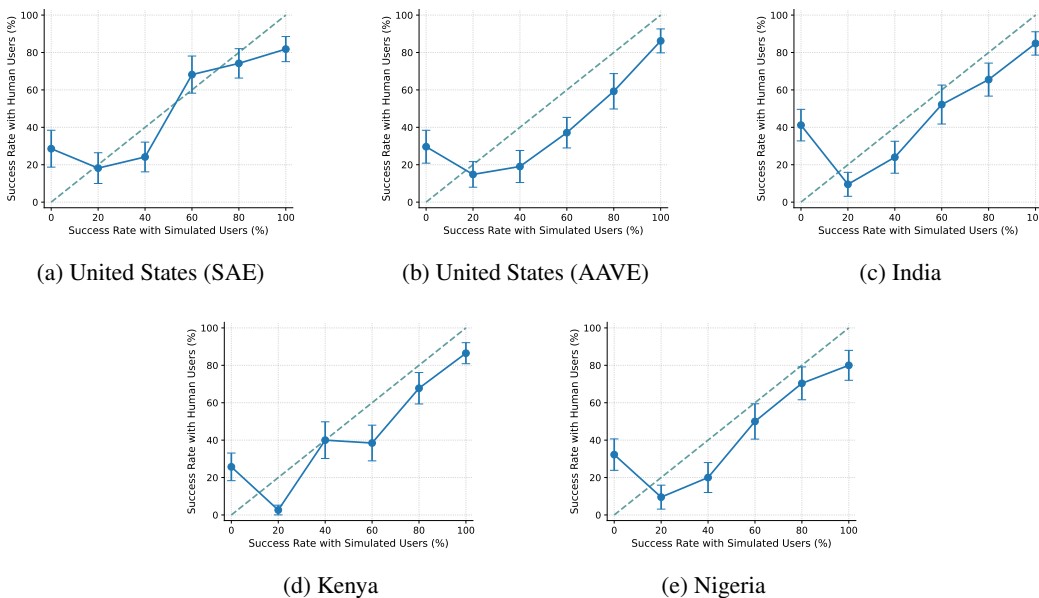

Figure 4: Success rate with human users vs. LLM-simulated users across dialect and country groups. All participants are in the 18–34 age group.

We observe contrasting age-related patterns in success rates across dialects. For SAE participants, success rates increase slightly with age ($\sim$3.0 percentage point increase from 18-34 to 55+ group), whereas for AAVE participants, success rates decrease with age (7.6 percentage point decrease from 18-34 to 55+ group). Notably, dialect disparities in agent performance grow larger with age: there is nearly a 12 percentage point decrease in agent performance between SAE and AAVE 35-54 groups and a 19 percentage point decrease in agent performance between SAE and AAVE 55+ groups (age-stratified GEE dialect effects: $\beta_{35-54} = 0.67$, $p = 0.01$; $\beta_{55+} = 1.24$, $p = 0.001$). The calibration gap is particularly pronounced for the 35-54 age group (10.3 $ECE_{\text{Human–LLM}}$ gap), where SAE primarily exhibits miscalibration for the 1st bin, while AAVE exhibits miscalibration across all bins (Figures 3b and 3e).

Note that success rate and $ECE_{\text{Human–LLM}}$ capture distinct aspects of performance. While *higher* success rate and *lower* $ECE_{\text{Human–LLM}}$ are both desirable, since higher success rates reflect stronger agent task performance and lower $ECE_{\text{Human–LLM}}$ indicates that simulated users serve as reliable proxies for human users, improvements in one do not necessarily imply improvements in the other. For example, SAE participants aged 18–34 exhibit both lower success rates and $ECE_{\text{Human–LLM}}$, whereas SAE participants aged 55+ exhibit both higher success rates and $ECE_{\text{Human–LLM}}$.

### 4.3.2 COUNTRIES

Due to participant availability constraints, we focus our cross-country analysis on the 18-34 age group. We find that differences in agent performance, shown in Table 3, are present but much less pronounced across countries (ranging from $41.0\%$-$49.2\%$) than those observed by dialect and age within the US (Table 2). In particular, Kenyan and Nigerian participants experience similar success rates ($43.5\%$ and $43.7\%$). A GEE model confirms that cross-country differences are *not* statistically significant (all $p > 0.49$).

| Group | Success | ECE |
|-------|---------|------|
| SAE | 49.2 | 13.0 |
| AAVE | 41.0 | 18.9 |
| India | 46.2 | 18.9 |
| Kenya | 43.5 | 15.6 |
| Nigeria | 43.7 | 17.6 |

Table 3: Success rate (%) and $ECE_{\text{Human–LLM}}$ across dialects and countries (18–34 age group).

Simulated users are best calibrated to SAE participants ($ECE_{\text{Human–LLM}} = 13.0$) and worst calibrated to AAVE and Indian participants ($ECE_{\text{Human–LLM}} = 18.9$), suggesting simulated users are especially poor proxies for these groups. Across countries, we observe that simulated users tend to exhibit strongest calibration for fairly to moderately difficult tasks (20%-40% difficulty bins). However, they

| Group | Arg (%) | Miss (%) | Extra (%) | Output (%) |
|---|---|---|---|---|
| **Simulated User** | | | | |
| – | 32.2 | 17.8 | 5.6 | 31.4 |
| **Human User – US, SAE** | | | | |
| All | 25.2 | 18.5 | 9.5 | 16.2 |
| 18–34 | 27.9 | 15.8 | 7.6 | 18.9 |
| 35–54 | 24.5 | 18.5 | 10.6 | 17.0 |
| 55+ | 23.2 | 21.3 | 10.3 | 12.2 |
| **Human User – US, AAVE** | | | | |
| All | 39.0 | 23.0 | 10.7 | 15.5 |
| 18–34 | 36.8 | 25.3 | 13.3 | 15.5 |
| 35–54 | 37.8 | 20.4 | 8.7 | 16.4 |
| 55+ | 45.8 | 24.1 | 9.6 | 13.3 |
| **Human User – Non-US, 18–34** | | | | |
| India | 33.3 | 18.2 | 7.9 | 20.4 |
| Kenya | 38.5 | 17.2 | 7.8 | 23.6 |
| Nigeria | 33.8 | 22.3 | 10.8 | 21.7 |

Table 4: Error breakdown (%) for simulated and human users, aggregated across tasks. The `min` and `max` are highlighted per column. Error types: **argument error** (action taken matches ground truth action but with different arguments), **missing action error** (ground truth action is missing from actions taken), **extra action error** (actions taken go beyond ground truth actions), and **output error** (expected outputs are missing/incorrect). Each error type is measured as a binary indicator per task (e.g., a missing action error records whether **any** required action is missing). Note that error percentages do not sum to 100%; some tasks have no errors while others have multiple error types.

consistently overestimate performance for easy tasks (80%-100% difficulty bins), shown in Figure 4. As a result, evaluations relying on simulated users risk systematically underestimating difficulty agents face when deployed to diverse, global user populations.

## 4.4 ANALYZING INTERACTIONS

### 4.4.1 STRUCTURE AND CONTENT

Interactions between agents and simulated users vs. human users follow similar structures and surface-level forms, including # of turns, # of actions, and # of words/turn. We observe some differences between simulated and human user interactions when considering conversational content (Table 6). Simulated user conversations include questions in 18.8% of user turns and 51.8% of agent turns, compared to 9.8% and 56.3%, respectively, for human users. Notably, Nigerian participants only ask questions in 4.3% of turns.

Differences are more pronounced when examining politeness indicators (e.g., please, thank you, apologize). Simulated user conversations include such indicators in 39.2% of user and 52.0% of agent turns, compared to 19.9% and 41.1%, respectively, for human users. For both question-asking and politeness indicators, differences in behavior are more pronounced between simulated and human users than among human users. We also show that targeted behavioral interventions, such as prompting simulated users to limit politeness, can alter calibration patterns and reduce gaps for highly miscalibrated groups (see Appendix A.7), indicating that prompting strategies can partially mitigate miscalibration issues.

### 4.4.2 ERRORS

We analyze differences in (1) error types and (2) error attribution for simulated vs. human user interactions to understand whether both groups experience the same failure modes. Error types include argument errors, missing actions, extra actions, and output errors (see Table 4 for definitions).

Agents make argument errors (performing the correct action with incorrect arguments) at rates of 32.2% for simulated users compared to 23.2%–45.8% for human users, with AAVE participants experiencing the highest errors. Agents tend to omit actions (17.8% vs. 15.8%–25.3%) or include

unnecessary ones (5.6% vs. 7.6%–13.3%) less frequently for simulated users compared to human users. However, output errors show the reverse pattern: agents either omit or include incorrect outputs more often for simulated users (31.4%) than for human users (12.2%–23.6%). These patterns suggest that agents exhibit different behavior when interacting with simulated vs. human users; they perform more complete and efficient action sequences but make more frequent output errors.

When comparing error attribution for simulated vs. human user conversations (Table 5), we observe clear differences in where task failures occur. For simulated conversations, agents are responsible for task failures substantially more often than for human conversations (48.9% vs. 24.5%). In contrast, users are the primary source of failure in human conversations (62.2% vs. 40%).

| Source | Simulated | Human |
|--------|-----------|-------|
| Agent  | 48.9      | 24.5  |
| User   | 40.0      | 62.2  |
| Both   | 2.2       | 11.1  |
| Other  | 8.9       | 2.2   |

Table 5: Error Attribution (%) for simulated vs. human user conversations with $reward = 0$, manually annotated by the authors ($n = 45$ per condition, matched on task difficulty).

These differences point to distinct failure patterns in simulated versus human interactions. The higher user error rate in human interactions reflects ambiguity, misunderstandings, or partial compliance that humans naturally introduce. Conversely, the higher agent error rate in simulated conversations suggests simulated users appear to exhibit more precise instruction following or adapt more readily to agent responses, placing greater burden on agents to execute correctly. In practice, this divergence may lead to misdiagnoses of failures. Simulation-based evaluations may overemphasize agent execution errors, while obscuring challenges that arise when real users engage with agents in ways not reflected by simulated users.

## 5 DISCUSSION AND CONCLUSION

As AI agents become integrated into everyday tasks, ensuring equitable performance across diverse populations is essential. However, the "user" component of agentic evaluations, central to real-world interaction, has largely been overlooked. Our findings reveal fundamental limitations of LLM-simulated users, showing that current simulation practices misestimate agent performance for actual users and obscure demographic disparities, which may result in evaluations optimizing for objectives that diverge from real-world use.

Systems optimized for simulated users may appear robust in benchmarks while failing disproportionately for real users in the wild, whose communication styles are underrepresented by simulation. For example, the behavioral artifacts we observe in simulated interactions—such as heightened politeness and question-asking—suggest that simulated users reflect communication norms that may not generalize across diverse user groups. Moving forward, agentic benchmarks should assess robustness across multiple simulation models, validate simulated outcomes against demographically diverse human data if possible, and transparently acknowledge the limitations of user simulation.

### LIMITATIONS

While our study provides important insights into user simulation in agentic evaluations, it is useful to clarify its scope and outline directions for future work. Our evaluation is conducted entirely in English (replicating a limitation of popular agentic benchmarks, which are limited to English), which restricts our ability to assess how LLM-simulated users behave in multilingual settings. Since language influences user and agent behavior, capabilities, and interaction norms, it is important to verify the extent to which our findings hold for non-English contexts. In addition, our age-based analyses are limited to users in the United States due to recruitment constraints, leaving open the question of how performance and calibration disparities vary with age across other countries and cultural contexts. Additionally, our analysis is limited to two common U.S. English dialects and does not capture the full range of dialectal variation within the United States. We also do not consider dialectal variation in non-U.S. contexts, where English usage reflects substantial regional and sociolinguistic diversity rather than a single uniform dialect.

We also evaluate agents in a single domain, focusing on retail customer service scenarios from $\tau$-Bench. While these tasks are designed to capture multi-turn, task-oriented interactions with tool

use, agent performance patterns and the quality of user simulation may differ in other domains such as healthcare, where interaction structure and task complexity vary. In such domains with longer-form interactions, individual style and cultural differences may be more apparent, potentially amplifying the effects we observe. Nevertheless, it would be important to empirically validate this expectation.

Finally, we focus on a single, fixed agent (GPT-4o). Holding the agent constant enables us to isolate the effects of user simulation on evaluation outcomes. However, we do not examine how calibration gaps or performance disparities vary across different agents, which is important for developing a more complete understanding of the robustness, validity, and fairness of user simulation. We discuss these points further in Appendix A.2.

## ACNKOWLEDGEMENTS

We thank Ava Batchkala, Boyu Fan, Nithya Govindarajan, Keith Hall, Mads Jenkins, Kelly Marchisio, Harry Moynehan, Kailash Saravanakumar, Alice Schoenauer Sebag, Priyanka Sen, Jimin Sun, and Pat Verga for piloting our user study and providing feedback. We also thank the members of UCI NLP for helpful discussions and comments. This work was conducted primarily during Preethi's internship at Cohere, and was supported in part by the Hasso Plattner Institute (HPI) and NSF CAREER award number IIS-2046873.

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

## A APPENDIX

### A.1 ETHICAL CONSIDERATIONS

All participants received standardized compensation that was not adjusted by country, ensuring consistent and fair payment regardless of geographic location. Participants were provided with an overview of the study procedures upfront and could withdraw from the study at any point without penalty. Participants provided informed consent by reading the study instructions and choosing to participate and complete the study. The study is classified as minimal risk, since it involves interaction with AI agents in simulated retail customer service scenarios and does not involve the collection of sensitive personal information.

Beyond the user study itself, our work has broader societal implications. Our results demonstrate that LLM-simulated users may not serve as reliable proxies for human users and can exhibit demographic biases. If simulated users are adopted as a standard practice for agent evaluation despite these limitations, there is a risk that AI systems could be deployed in ways that systematically underserve certain demographic groups. We hope this work encourages the research community to reflect on current evaluation practices and to develop agentic evaluation approaches that better represent diverse user populations.

### A.2 DESIGN CHOICES

**Single Agent** We use GPT-4o as the agent model throughout all experiments to isolate the effect of user variation while holding agent capability constant. This design choice allows us to attribute observed differences in performance to user simulation rather than differences in agent performance.

| Group | Turns | Actions | W/T (U) | W/T (A) | Q (U) | Q (A) | P (U) | P (A) |
|---|---|---|---|---|---|---|---|---|
| **Simulated User** | | | | | | | | |
| – | 16.2 | 7.9 | 13.4 | 55.0 | 18.8 | 51.8 | 39.2 | 52.0 |
| **Human User – US, SAE** | | | | | | | | |
| All | 15.0 | 7.4 | 12.6 | 53.0 | 11.6 | 56.6 | 19.1 | 41.1 |
| 18–34 | 15.8 | 7.6 | 11.4 | 55.4 | 14.5 | 58.1 | 15.6 | 41.1 |
| 35–54 | 14.9 | 7.3 | 11.6 | 51.5 | 9.8 | 55.3 | 19.7 | 43.6 |
| 55+ | 14.3 | 7.3 | 14.8 | 51.9 | 10.3 | 56.2 | 22.1 | 38.7 |
| **Human User – US, AAVE** | | | | | | | | |
| All | 14.9 | 7.4 | 12.4 | 52.8 | 10.0 | 56.4 | 19.7 | 42.5 |
| 18–34 | 15.5 | 7.6 | 11.9 | 52.6 | 12.8 | 56.8 | 17.4 | 44.9 |
| 35–54 | 14.8 | 7.3 | 12.1 | 53.4 | 8.9 | 57.2 | 19.9 | 42.1 |
| 55+ | 15.3 | 7.0 | 13.9 | 51.9 | 9.1 | 53.9 | 24.0 | 38.3 |
| **Human User – Non-US, 18–34** | | | | | | | | |
| India | 16.3 | 7.4 | 11.5 | 53.1 | 9.9 | 57.7 | 20.1 | 40.8 |
| Kenya | 14.4 | 8.0 | 13.7 | 54.6 | 8.9 | 54.7 | 21.4 | 39.1 |
| Nigeria | 14.4 | 7.5 | 11.2 | 55.6 | 4.3 | 56.8 | 18.7 | 41.4 |

Table 6: Conversational statistics (means) for simulated users and human users (split by demographic group). Statistics include: **Turns** – number of turns in the interaction, **Actions** – total number of actions performed by the agent (including read and write actions), **W/T (U/A)** – words per turn, for the user/agent, **Q (U/A)** – percent of turns with a question, for the user/agent, **P (U/A)** – percent of turns with politeness indicators (e.g., please, thank you, apologize, etc.), for the user/agent.

While agent model choice may influence absolute success rates, the core questions investigated in this paper (robustness, validity, and fairness of user simulation) are agent-agnostic.

We acknowledge that we cannot assess whether these issues vary across agents of different capabilities. Future work should examine whether weaker or stronger agents exhibit different calibration patterns when interacting with simulated vs. human users.

**Single Benchmark** Our analysis focuses on $\tau$-Bench as a case study, but we expect similar evaluation concerns to arise across agentic benchmarks that rely on simulated users. The core challenges we identify (sensitivity to user simulation model, miscalibration across difficulty levels, and demographic biases) are likely to apply to AI agents designed to perform multi-turn, tool-using task completion. However, benchmarks with different task structures (e.g., web navigation vs. conversational assistance), evaluation metrics (e.g., trajectory-based vs. outcome-based), or user simulation prompting strategies may exhibit varying degrees of these issues. A comprehensive investigation across multiple benchmarks remains an important direction for future work.

## A.3 DIALECT SCREENING

For US participants, we used Prolific's participant filters to recruit White and Black participants. As part of prescreening (in addition to questions about education, AI experience, and AI usage), we asked participants to self-identify their primary English dialect. Specifically, we asked: "Which of the following best describes your everyday English usage?" with the following response options: (1) Mostly Standard American English (SAE), (2) Mostly African American Vernacular English (AAVE), (3) A mixture of SAE and AAVE, and (4) Not sure. We retained participants who identified as White/SAE or Black/AAVE for our analysis.

## A.4 STATISTICAL ANALYSIS

We use Generalized Estimating Equations (GEE) to assess the statistical significance of demographic differences in task success while accounting for the repeated-measures structure of our data, as each participant completes four tasks. We model binary success outcomes using a binomial family and cluster observations by participant ID.

We fit two types of models: (1) an overall model including all participants with covariates for age (for United States only), dialect/country, education, AI exposure, AI usage, and task difficulty

Figure 5: User study chat interface where participants interact with the agent (GPT-4o) to complete tasks.

(operationalized via model-based success-rate bins described in Section 3.2) and (2) age-stratified models fit separately for each age group (18–34, 35–54, 55+) to examine how dialect effects vary by age. Coefficients represent log-odds of task success; we report estimated coefficients and associated p-values.

## A.5 $\tau$-BENCH ADAPTATION

We modify task instructions to ensure that neither simulated nor human users are influenced by identity and behavioral cues in the instructions when completing tasks. First, we remove user names (e.g., Yusuf Rossi) from instructions and replace them with anonymized user IDs following the format [a-z][0-9][0-9] (e.g., o32). Second, we remove behavioral cues (e.g., "You are detail-oriented and want to make sure everything is addressed in one go") to avoid biasing user behavior (Tseng et al., 2024; Xing et al., 2025). These modifications preserve all task objectives and requirements.

**Example of Original Task Instructions**   *You are Yusuf Rossi in 19122. You received your order #W2378156 and wish to exchange the mechanical keyboard for a similar one but with clicky switches and the smart thermostat for one compatible with Google Home instead of Apple HomeKit. If there is no keyboard that is clicky, RGB backlight, full size, you'd rather only exchange the thermostat. You are detail-oriented and want to make sure everything is addressed in one go.*

**Example of Adapted Task Instructions**   *You are User b63 in 19122. You received your order #W2378156 and wish to exchange the mechanical keyboard for a similar one but with clicky switches and the smart thermostat for one compatible with Google Home instead of Apple HomeKit. If there is no keyboard that is clicky, RGB backlight, full size, you would rather exchange only the thermostat. You want to make sure everything is addressed in one go. To start the conversation, say 'Hello, my email is user.b63@example.com.'*

## A.6 USER STUDY INTERFACE AND INSTRUCTIONS

Users interact with a Streamlit app (Figure 5) to converse with the agent and complete tasks. We provide participants with the following general instructions, which apply to all tasks in addition to the task-specific instructions shown in Table 8.

| Age Group | $\Delta ECE$ |
|---|---|
| **US, SAE** | |
| All | 2.4 |
| 18–34 | -5.0 |
| 35–54 | 3.3 |
| 55+ | 5.5 |
| **US, AAVE** | |
| All | -4.9 |
| 18–34 | -2.6 |
| 35–54 | -6.0 |
| 55+ | -0.3 |
| **Non-US, 18–34** | |
| India | -6.2 |
| Kenya | -4.2 |
| Nigeria | 1.0 |

Table 7: Differences between $ECE_{\text{Human–LLM}}$ for reduced-politeness user simulation across demographic groups vs. standard user simulation. Negative values indicate better calibration after behavioral intervention. Human user study results are held fixed, while task difficulty bins are recomputed based on updated simulated user results, resulting in changes to $ECE_{\text{Human–LLM}}$.

**Instructions:** Please respond as the user described in the task instructions. You want to complete all the requests mentioned in the instructions. The agent is there to assist you with completing the task. Do not make up information beyond what the instructions provide. You can tell the agent you are unsure and ask them to look up information based on your profile or orders. Beyond this, please behave naturally and converse as you normally would. Use the 'End Conversation' button in the left sidebar to finish your conversation. To begin the conversation, authenticate yourself by providing the user email provided in the instructions.

Note that all task instructions are free of user persona information and use generic user IDs to avoid biasing interactions.

## A.7 Behaviorally Targeted User Simulation

We previously identified heightened politeness as a behavioral artifact in simulated user interactions. Motivated by this observation, we examine whether explicitly constraining this behavior affects evaluation outcomes. To test this, we introduce a minimal intervention to the user simulation prompt that limits the use of politeness indicators while preserving task objectives: "You may include politeness indicators (e.g., please, thank you, sorry) occasionally, but use them sparingly. Limit their use to at most one or two times across the entire conversation." We re-run user simulation on the 18-task subset using GPT-4o as both the agent and user LLMs, and re-bucket tasks based on the updated performance distribution. We find that 11 of 18 tasks shift difficulty bins, indicating that task difficulty estimates are sensitive to behavioral prompting, and overall success decreases from 50.0% to 46.7%.

Comparing $ECE_{\text{Human–LLM}}$ with and without behavioral prompting, we observe improved calibration for AAVE, Indian, and Kenyan participants—groups that previously exhibited the largest calibration errors—while calibration worsens for SAE and Nigerian participants (Table 7). These results highlight the sensitivity of user simulation to prompt-level choices and suggest that targeted behavioral interventions can alter calibration patterns.

## A.8 Country-Based User Simulation

In addition to the behavioral intervention, we also perform a demographic prompting intervention by providing a country-specific first name and location indicator (e.g., "Your name is Ramesh and you are from Mumbai, India.") to the user simulation model. By doing so, we examine whether explicitly providing demographic indicators to the model results in simulated users being more aligned with real users from a given country, relative to the default simulation setup. For each country, we ask ChatGPT to generate typical female and male names (9 each) that are frequently used and country-specific,

| Bin | Success | Example Task |
|-----|---------|--------------|
| 1 | 0% | Your name is User e70 and your zip code is 32190. You just bought a water bottle with 500ml but you regret it, and you instead want to change it to the other bottle you recently ordered with 1000ml capacity. If the exact 1000ml bottle is not available any more, you can allow the material to be different. To start the conversation, say 'Hello, my email is user.e70@example.com.' |
| 2 | 20% | You are User l64, and you live in Denver, 80280. You just won a lottery, and you want to upgrade all your items to the most expensive options (but make sure the shoe is still the same size). You want to pay the difference with your gift card, but if it is impossible, PayPal is fine. To start the conversation, say 'Hello, my email is user.l64@example.com.' |
| 3 | 40% | Your name is User u52, and you live in 46236. Your email is user.u52@example.com. You just placed an order but you realize that your card has only $1131 credit left, and the order total is more than $1160. You wonder if the agent can help split the payment with another card. If not, you wonder what the most expensive item is and its price, and if you can just cancel that item. If not, you wonder if you can switch all items to their cheapest options and bring the cost down to $1131. If so, do it. If not, you wonder if the agent can just cancel the order so that you can order again. To start the conversation, say 'Hello, my email is user.u52@example.com.' |
| 4 | 60% | You are User i49, and you live in 32286. You want to exchange your skateboard for a shorter bamboo material one. If several options are available, you want to know all options and their prices, and choose the most expensive one because you believe price is quality. Also, you want to exchange the garden hose you received to the type that you just ordered (pending). To start the conversation, say 'Hello, my email is user.i49@example.com.' |
| 5 | 80% | You are User b63 in 19122. You received your order #W2378156 and wish to exchange the mechanical keyboard for a similar one but with clicky switches and the smart thermostat for one compatible with Google Home instead of Apple HomeKit. If there is no keyboard that is clicky, RGB backlight, full size, you would rather exchange only the thermostat. You want to make sure everything is addressed in one go. To start the conversation, say 'Hello, my email is user.b63@example.com.' |
| 6 | 100% | You are User p59, residing in Philadelphia 19031. You want to change the Desk Lamp in order #W9300146 that you've placed for the cheapest Desk Lamp that's available. Any price difference should go to a gift card. You also want to know how much you get back in total. To start the conversation, say 'Hello, my email is user.p59@example.com.' |

Table 8: Example $\tau$-Bench tasks across difficulty bins. Task difficulty is determined by the percentage of evaluation runs (out of five) in which the agent succeeds when interacting with simulated users for a given task..

avoiding names common across multiple countries. For location, we select a major metropolitan city in each country: Mumbai for India, Nairobi for Kenya, and Lagos for Nigeria. Using the same procedure as the previous analysis, we re-run user simulation with the updated prompts and re-bucket tasks.

We observe some differences in agent performance depending on the country used to simulate users across 5 runs: $55.6 \pm 6.1\%$ for India, $51.1 \pm 6.5\%$ for Kenya, and $47.8 \pm 4.4\%$ for Nigeria. However, there are no apparent stereotypical linguistic markers or cultural references among interactions from different countries, suggesting that these differences may stem from more subtle or implicit cues.

Additionally, we see differences in calibration with and without demographic prompting ($\Delta ECE_{\text{Human–LLM}}$), with negative values indicating better calibration after the intervention. Calibration improves for Indian participants ($-7.9$), remains largely unchanged for Nigerian participants ($0.5$), and worsens for Kenyan participants ($8.6$). These results suggest that explicitly specifying user identity can influence calibration between simulated and real users in both desirable and undesirable ways, which is consistent with prior work showing that demographic prompting can have mixed effects (Durmus et al., 2024; Sun et al., 2025). However, more work is needed to comprehensively investigate the effectiveness and sensitivity of demographic prompting in agentic evaluations.

