# OpenReview forum: "Lost in Simulation: LLM-Simulated Users are Unreliable Proxies for Human Users in Agentic Evaluations"
_ICLR.cc/2026/Workshop/AFAA — AFAA 2026 Oral_

### Official Review · Reviewer_Rn2g · 2026-02-09
**Good but Isolated Case Study**

**Rating:** 4
**Confidence:** 3

**Summary:**

The paper presents a case study on τ-Bench retail, replacing the standard LLM-simulated user with human participants. It first runs GPT-4o as both agent and user across the benchmark and uses the resulting success rates to define six discrete difficulty bins; from these it selects 18 tasks (3 per bin) to cover difficulty evenly. The authors then recruit ~40 participants per group from the United States, India, Kenya, and Nigeria, with U.S. participants split into White SAE and Black AAVE speakers (and further stratified by age). Participants complete tasks by chatting with a fixed GPT-4o agent, and the resulting human success rates are compared against simulator-based evaluation. They find that simulated-user results are not a reliable proxy for humans: performance differs across populations, with lower success for AAVE users and generally worse outcomes for non-U.S. participants than simulator-based evaluations suggest.

**Strengths:**

I believe this research direction is valuable and the hypotheses are interesting. The experimental setup evaluates them well within the scope selected by the manuscript (see weaknesses for some issues, though).

Sourcing the subjects looks correct to me, and I believe the observed effects isolate what they want to measure. The care taken to ensure this is accurate constitutes real merit.

I like that they included an experiment prompting GPT-4o to behave like specific personas in Appendix A.7-A.8. This could be referenced more prominently in the main body.

Section 4.4 analyzing the interactions is interesting, and I am glad the authors included it.

The paper provides concrete recommendations that would be useful for any benchmark using simulated users: test across multiple user models, validate with human data, and report limitations of simulation.

While I discuss the weaknesses below, I believe the manuscript is still useful for the community and I am leaning towards acceptance.

**Weaknesses:**

The choice of τ-bench as the case study could be better motivated. It is not obvious to me that τ-bench is representative of benchmarks that explicitly aim for human-faithful user simulation. If τ-bench was designed primarily for scalable automated evaluation rather than faithfully simulating human users, it is unclear whether the findings generalize to benchmarks where user simulation has been specifically optimized for realism. The paper would be strengthened by discussing how representative τ-bench is of current user simulation practices.

The operationalization of task difficulty is problematic. Difficulty is defined by GPT-4o's success rate with simulated users, but the consistent U-curve pattern across Figures 2-4 suggests the lowest bin captures systematic model failures rather than inherently difficult tasks. Human success at the 0% bin actually exceeds success at intermediate bins, which is not what we would expect if difficulty were monotonic. This undermines the claim that simulated users "underestimate agent performance on challenging tasks." I would want to see either an independent validation of difficulty labels (e.g., annotation of task complexity, number of required actions) or explicit acknowledgment of this limitation with adjusted conclusions.

This is also generally tied to the fact that the paper does not cleanly separate agent capability from user simulation fidelity. When we observe calibration gaps, it is unclear how much stems from unfaithful simulation versus agent-specific failures with certain communication styles versus model-specific interactions when GPT-4o serves as both agent and user. While probably not needed for a workshop paper, more controlled design, perhaps using multiple agents or matched message comparisons, would help disentangle these factors and make these results more generalizable.

---

### Official Review · Reviewer_oRen · 2026-02-14
**Lost in Simulation: LLM-Simulated Users are Unreliable Proxies for Human Users in Agentic Evaluations**

**Rating:** 4
**Confidence:** 4

**Summary:**

This study investigates the validity of using LLMs as proxies for human users in agentic benchmarks, specifically within the retail domain. By comparing agent performance across a diverse human cohort against LLM simulations, the authors identify three critical findings: agent success rates shift by roughly 9pp based on the simulator used, simulated evaluations are poorly calibrated and performance gaps are disproportionately wider for AAVE speakers, particularly among older demographics. The findings suggest that relying on simulation can lead to inaccurate performance projections and the erasure of demographic disparities.

**Strengths:**

The following are the strengths:
1) The paper provides a solid comparison between simulated agents and real world users - the basis and results of the paper are practical and not theoretical.
2) There is a good representation of people from different demographic regions. The dialects and age are analyzed in detail and there is good statistical analysis done to emphasize the results.
3) The use of the ECE metric is a great way to formalize miscalibration.

**Weaknesses:**

The following are the weaknesses that can be addressed:
1) A single benchmark for retail taks is studied, it would be interesting to see how the claims generalize to other tasks.
2) GPT-4o is used both as the judge and also as the simulation agent  hence making the difficulty scale used in the analysis not purely objective.

---

### Official Review · Reviewer_YVZr · 2026-02-18
**Review: Lost in Simulation — LLM-Simulated Users are Unreliable Proxies for Human Users in Agentic Evaluations**

**Rating:** 4
**Confidence:** 4

**Summary:**

This paper investigates whether LLM-simulated users serve as reliable proxies for real human users in agentic benchmarks, using τ-Bench retail tasks as a case study. Through a user study with 240+ participants from the US, India, Kenya, and Nigeria, the authors evaluate robustness (consistency across user LLMs), validity (calibration between simulated and human performance), and fairness (demographic disparities). Key findings: (1) agent success rates vary up to 9pp across different user simulation models; (2) simulated users exhibit systematic miscalibration (ECE_Human-LLM = 15.1 for US participants); (3) AAVE speakers experience 11.2pp lower success rates and 8.6pp higher calibration error than SAE speakers, with disparities compounding with age; (4) simulated users introduce conversational artifacts (heightened politeness, question-asking) and surface different error patterns than humans.

**Strengths:**

Strengths
1. Timely and important research question. Agentic benchmarks widely adopt user simulation without validation against real users. This paper directly addresses a critical gap in evaluation methodology that has practical implications for agent deployment.
2. Well-designed empirical study with demographic diversity. The recruitment across 4 countries, stratification by age and dialect (SAE/AAVE), and collection of ~240 participants × 4 tasks provides substantial coverage. The focus on AAVE speakers is particularly valuable given their underrepresentation in AI fairness research applied to interactive systems.
3. Multi-faceted evaluation framework. The paper examines robustness, validity, and fairness as distinct dimensions, each with clear operationalizations (variance across user LLMs, ECE calibration metric, demographic stratification). The adaptation of ECE from confidence calibration to simulation calibration is conceptually sound.
4. Rigorous statistical analysis. Use of GEE models to account for repeated measures, explicit reporting of confidence intervals, and age-stratified analyses strengthen the reliability of demographic findings. The statistical significance testing (β = 0.61, p < 0.001 for dialect effect) is appropriate.
5. Actionable insights. The behavioral intervention (Appendix A.7) showing that reduced-politeness prompting alters calibration patterns demonstrates that prompt-level choices matter, providing a practical avenue for mitigation.
6. Transparent limitations. The paper explicitly acknowledges scope constraints (English-only, single domain, single agent, no multi-agent analysis) and avoids overclaiming.

**Weaknesses:**

Weaknesses
1. Single benchmark, single agent limits generalizability. All findings are based on τ-Bench retail tasks with GPT-4o as the agent. The paper argues these are "agent-agnostic" concerns (Appendix A.2), but this is unverified. Key questions:

Do weaker agents (e.g., older models, open-source) show similar calibration gaps?
Do other agentic benchmarks (WebArena, AgentCompany) exhibit the same demographic disparities?
Are the observed artifacts (politeness, question-asking) specific to customer service domains or generalizable?

Without multi-benchmark or multi-agent validation, it's unclear whether the 11.2pp SAE-AAVE gap or 9pp cross-LLM variance are fundamental issues with user simulation or τ-Bench-specific phenomena.

2. ECE metric conflates multiple sources of miscalibration. ECE_Human-LLM measures weighted absolute deviation across difficulty bins, but does not decompose why calibration fails. Possible sources include:

Task difficulty estimation error (binning by simulated performance may not reflect human difficulty)
Agent behavior changes when interacting with different user types
Evaluation metric artifacts (substring matching for output validation)

The paper attributes miscalibration to "simulated users differing from real users" but does not isolate this from agent-side or evaluation-side confounds. A cleaner test would hold evaluation constant and vary only the user.

3. Demographic prompting results (Appendix A.8) are underexplored. The finding that country-specific demographic prompting improves calibration for India (∆ECE = -7.9) but worsens it for Kenya (+8.6) is striking but receives minimal analysis. This suggests:

Stereotypical name/location cues may help or harm depending on how well they match LLM training distributions
The mechanism is unclear — is this about linguistic priors, cultural associations, or spurious correlations?

Given the paper's emphasis on fairness, this deserves a deeper investigation or at least a clearer discussion of when demographic prompting helps vs. harms.

4. Error attribution analysis (Table 5) is based on small, manually annotated sample. Only 45 simulated and 45 human conversations are annotated for error attribution, and there is no inter-annotator agreement reported. The claim that "agents are responsible for failures 48.9% of the time with simulated users vs. 24.5% with humans" is a strong finding, but:

No confidence intervals or statistical tests are provided
The manual annotation procedure is not described (who annotated? what were the guidelines?)
45 samples may be insufficient given the diversity of failure modes

This weakens one of the paper's key mechanistic claims about why simulated and human interactions differ.

---

### Meta-Review · Area_Chair_LZti · 2026-02-19

**Recommendation:** Main Papers Track
**Confidence:** 4

**Metareview:**

The reviewers agree on the relevance of the paper for the workshop and the timeliness of the study. However, some concerns persist across the reviews, including 1) details about the annotations 2) the use of a single model. Nonetheless, the paper can raise interesting discussions with the community about the evaluations of agents and thus I believe it should be accepted.

---

### Decision · Program_Chairs · 2026-03-02

Accept (Oral)